# Moral Responsibility for AI Systems

**Sander Beckers**
University of Amsterdam
sanderbeckers.com
srekcebrednas@gmail.com

## Abstract

As more and more decisions that have a significant ethical dimension are being outsourced to AI systems, it is important to have a definition of *moral responsibility* that can be applied to AI systems. Moral responsibility for an outcome of an agent who performs some action is commonly taken to involve both a *causal condition* and an *epistemic condition*: the action should cause the outcome, and the agent should have been aware – in some form or other – of the possible moral consequences of their action. This paper presents a formal definition of both conditions within the framework of causal models. I compare my approach to the existing approaches of Braham and van Hees (BvH) and of Halpern and Kleiman-Weiner (HK). I then generalize my definition into a *degree of responsibility*.

## 1 Introduction

As more and more decisions that have a significant ethical dimension are being outsourced to AI systems, it is important to have a definition of responsibility that can be *applied* to the decisions of AI systems, and that can be *used by* AI systems in the process of its decision-making [8]. To meet the first condition, such a definition should require only a minimal notion of agency and instead focus on those aspects of responsibility that are readily applicable to (current) AI systems. To meet the second condition, such a definition should be formulated in a language that can be implemented into an AI system, so that it can integrate judgments of responsibility into its decision-making. This paper sets out to propose such a definition using the well-established framework of causal models [22, 23].

There exist different notions of moral responsibility that one might be interested in, and here we restrict attention to just one of them, namely *responsibility for consequences*, meaning the responsibility one has for a particular outcome that is the result of performing a particular action. This can be expressed more clearly by saying that the action *caused* the outcome, and therefore the first condition of concern here is the **causal condition** on responsibility [12, 26, 21, 5]. The past two decades have seen immense progress on offering formal definitions of actual causation by way of using causal models, and the definition here developed takes maximal advantage of this progress by comparing some recent proposals and choosing the one that correctly handles several complicated cases to be considered [33, 16, 2].

Our actions can cause all kinds of outcomes for which we are clearly not morally responsible: if a train crashes into a car that illegally crosses the railroad then the train conductor is not responsible for the car driver's death, if you turn on a light switch in a hotel room then you are not responsible if a short-circuit follows, etc.. The standard intuition that we have in such cases is that the agent "could not have known" that their action would cause the outcome. This is why definitions of responsibility also invoke an **epistemic condition**, stating roughly that the agent should have been able to foresee that they are performing an action which could result in them being responsible for the outcome [5, 27, 25].

37th Conference on Neural Information Processing Systems (NeurIPS 2023).

In addition to the causal and the epistemic conditions, it is standard to demand that responsibility also requires the fulfilment of a **control condition** (sometimes also called freedom condition), which expresses the fact that the agent had the right sort of control whilst performing their action [25]. Due to its close connection to issues of free will and determinism, this condition is heavily debated within philosophy. Within the context of (current) AI systems, however, the control condition can take on a more mundane form: any action that was a result of the correct operation of its program can be viewed as being under the AI's control. Therefore I simply take there to be a specific action variable that ranges over a set of possible actions, and assume that whenever the AI system is running successfully it has control over the value that this variable takes.

My approach proceeds along the same lines as that of Braham and van Hees (BvH) [5]. They offer the most influential formalization of moral responsibility that incorporates both the causal and the epistemic conditions, and therefore their work forms an appropriate point of comparison. Although I agree with the spirit of their approach, I disagree with its formulation. First, their causal condition defines causation as being a *Necessary Element of a Sufficient Set* (NESS). However, their use of game-theory instead of causal models results in an overly simplistic view of NESS-causation that cannot handle indirect causation. Therefore I first formulate their definition using causal models, and then show how to modify it so that it can overcome this limitation. Second, I disagree with the particulars of both their causal and their epistemic conditions. I argue for replacing the NESS definition of causation with my recently developed *Counterfactual NESS* (CNESS) definition [2]. Their epistemic condition states that the agent should minimize the probability of causation. I argue for giving that condition a secondary role: minimizing the probability of causing the outcome is subservient to minimizing the probability of the outcome simpliciter. I analyze several examples to illustrate the superiority of my conditions.

More recently, Halpern & Kleiman-Weiner (HK) [17] used causal models to propose definitions of several concepts that are closely related to moral responsibility. Although they do not explicitly define moral responsibility, they do suggest using the modified Halpern & Pearl (HP) definition of causation for the causal condition [16]. The HP definition correctly handles most of the counterexamples to the NESS definition here presented, but I discuss two types of example for which it fails (whereas the CNESS definition does not). HK also offer a definition of "degree of blameworthiness" that for all intents and purposes is very similar to an epistemic condition: it measures the extent to which the agent minimized the probability of the outcome. I present a case in which the epistemic conditions of BvH and HK conflict in order to argue that a more elaborate epistemic condition is required. My epistemic condition combines that of HK with that of BvH by demanding that an agent minimizes the probability of the outcome, but if possible also minimizes the probability of causation. [1]

Here is the general schema that encompasses all definitions of responsibility that I aim to consider.

**Responsibility Schema.** *An agent who performs $A = a$ is responsible for outcome $O = o$ if:*

- **(Control Condition)** *The agent had control over $A = a$.*
- **(Causal Condition)** $A = a$ causes $O = o$.
- **(Epistemic Condition)** *The agent believes that they could have avoided being responsible for $O = o$ by performing some alternative action $A = a'$.*

As mentioned, I simply assume that the **Control Condition** is always met (as do BvH, who call it the Agency Condition). For sake of brevity, I leave it implicit from now on.

Formalizing the **Causal Condition** comes down to settling the discussion on how to formalize actual causation, which has received considerable attention over the past two decades [22, 31, 15, 30, 16, 1]. A full discussion of causation would be too ambitious for the present purposes. Instead, I evaluate the suitability of several definitions of causation within the context of responsibility by presenting examples that bring across how they differ. On the basis of this evaluation I suggest adopting the CNESS definition and refer the reader to [2] for a more general motivation.

The **Epistemic Condition** requires settling the question: what does it take for the agent to believe that performing $A = a'$ allows them to avoid responsibility? Since this condition uses the notion of

---

[1]Note that the epistemic conditions of HK and BvH are not necessarily inconsistent: if one simply defines causation as an increase in the probability of the outcome occurring, they become equivalent. Except for the fact that he uses objective probabilities rather than those of the agent, this is roughly the proposal defended in [29]. As exemplified by the examples to be discussed (and as exemplified by browsing the recent literature on causation) such a naive probabilistic approach to causation is unable to deliver sensible verdicts.

responsibility, our Schema is circular. There exist different ways of filling it in so that it no longer is circular, and it is this flexibility that makes filling in the condition interesting. One possible suggestion is to demand that the agent believed $A = a'$ would not result in the outcome $O = o$, another weaker suggestion is to demand that the agent believed $A = a'$ would not cause $O = o$, etc..

A note of clarification is in order before we proceed. The current work does not aim to offer a complete theory of moral responsibility for AI systems, but rather zooms in on the above conditions whilst ignoring certain others. Concretely, here are some important issues that I set aside in this paper.

## 1.1 Some Limitations and Related Work

There exist forms of moral responsibility that do not (always) involve causation, such as those that follow from certain societal norms and expectations. For example, a captain is responsible for everything that happens on their ship, a parent is responsible for the behavior of their child, etc.. More generally, assigning responsibility to AI systems should itself be seen as just one part of the wider discussion on accountability that arises from the introduction of such systems into our society [19, 7].

Relatedly, responsibility is often associated with the morally stronger notions of blame and praise. I take responsibility to be a weaker notion that necessarily precedes judgments of blame and praise: one cannot be blameworthy for an outcome unless one is responsible for it, and similarly for praise. To develop definitions of blame and praise requires bringing into view both the *absolute* moral valence of the outcome $O = o$ (was it good or bad?) and its *relative* valence (was it better/worse than an alternative which it prevented?), as well as the costs incurred by the agent when performing an action. As the vast literature on trolley cases and other moral dilemmas illustrates, these issues make matters significantly more complex [10, 20].

One condition in particular that seems highly relevant to assigning blame (resp. praise) is to consider whether the outcome caused by the agent is *harmful* (resp. beneficial) or not. Indeed, one natural way of implementing a formal definition of responsibility within AI systems is to demand that it tries to avoid becoming responsible for harmful outcomes. This is confirmed by the recent European AI Act, which categorizes the risk that an AI system poses based on how likely it is to cause harm [11]. Beckers et. al. recently proposed a causal analysis of harm that is also formalized using causal models, and thus it could easily be integrated into the present proposal [3, 4].[2] In the present paper, however, I choose to focus exclusively on defining responsibility, thereby paving the way for future definitions of blame and praise.

Duijf presents a formalization of moral responsibility for outcomes that is likewise inspired by, but not an endorsement of, BvH [9]. Rather than defending an alternative definition of responsibility as I do, he presents a broad lanscape of completely formal conditions for responsibility that one might consider and analyzes their logical relations. As with BvH, his definition of NESS causation is formulated using game-theory, and thus it is likewise restricted to applications of direct causation.

The next section introduces the formalism of causal models that will be used to express all candidate definitions and related notions. Section 3 presents the BvH and HK definitions of responsibility and their respective definitions of causation. Section 4 discusses the **Causal Condition** by introducing two more definitions of causation and offers some examples to argue in favor of adopting the CNESS definition. (Some further examples are offered in the appendix.) We move on to a discussion of the **Epistemic Condition** in Section 5, which leads the way to my definition of moral responsibility. Since responsibility is often taken to come in degrees, in Section 6 I define the *degree of responsibility* and sketch how it helps interpret recent empirical work in psychology on responsibility judgments.

## 2 Causal Models

This section reviews the definition of causal models as understood in the structural modeling tradition started by Pearl [22], where I use the notation from Halpern [16].

**Definition 1.** *A signature $\mathcal{S}$ is a tuple $(\mathcal{U}, \mathcal{V}, \mathcal{R})$, where $\mathcal{U}$ is a set of* exogenous *variables, $\mathcal{V}$ is a set of* endogenous *variables, and $\mathcal{R}$ a function that associates with every variable $Y \in \mathcal{U} \cup \mathcal{V}$ a nonempty*

---

[2]I should note that they use the HP-definition of causation, which I criticize below. However, they state explicitly that their approach applies just as well to other definitions of causation.

*set $\mathcal{R}(Y)$ of possible values for $Y$ (i.e., the set of values over which $Y$ ranges). If $\vec{X} = (X_1, \ldots, X_n)$, $\mathcal{R}(\vec{X})$ denotes the crossproduct $\mathcal{R}(X_1) \times \cdots \times \mathcal{R}(X_n)$.*

Exogenous variables represent unobserved factors whose causal origins are outside the scope of the causal model, such as background conditions and noise. The values of the endogenous variables, on the other hand, are causally determined by other variables within the model.

**Definition 2.** *A causal model $M$ is a pair $(\mathcal{S}, \mathcal{F})$, where $\mathcal{S}$ is a signature and $\mathcal{F}$ defines a function that associates with each endogenous variable $Y$ a structural equation $F_Y$ giving the value of $Y$ in terms of the values of other endogenous and exogenous variables. Formally, the equation $F_Y$ maps $\mathcal{R}(\mathcal{U} \cup \mathcal{V} - \{Y\})$ to $\mathcal{R}(Y)$, so $F_Y$ determines the value of $Y$, given the values of all the other variables in $\mathcal{U} \cup \mathcal{V}$.*

We usually write the equation for an endogenous variable as $Y = f(\vec{X})$, where $\vec{X}$ are called the *parents* of $Y$ (and $Y$ is called a *child* of each variable in $\vec{X}$), and the function $f$ is such that it only depends on the values of $\vec{X}$. The *ancestor* relation is the transitive closure of the parent relation. In this paper we restrict attention to *acyclic* models, that is, models where no variable is an ancestor of itself. A (directed) *path* is a sequence of variables in which each element is a child of the previous element. In this manner an acyclic causal model induces a unique *DAG*, i.e., a Directed Acyclic Graph, which is simply a graphical representation of all the ancestral relations.

An *intervention* has the form $\vec{X} \leftarrow \vec{x}$, where $\vec{X}$ is a set of endogenous variables. Intuitively, this means that the values of the variables in $\vec{X}$ are set to the values $\vec{x}$. The equations define what happens in the presence of interventions. The intervention $\vec{X} \leftarrow \vec{x}$ in a causal model $M = (\mathcal{S}, \mathcal{F})$ results in a new causal model, denoted $M_{\vec{X} \leftarrow \vec{x}}$, which is identical to $M$, except that $\mathcal{F}$ is replaced by $\mathcal{F}^{\vec{X} \leftarrow \vec{x}}$: for each variable $Y \notin \vec{X}$, $F_Y^{\vec{X} \leftarrow \vec{x}} = F_Y$ (i.e., the equation for $Y$ is unchanged), while for each $X'$ in $\vec{X}$, the equation $F_{X'}$ for $X'$ is replaced by $X' = x'$ (where $x'$ is the value in $\vec{x}$ corresponding to $X'$).

Given a signature $\mathcal{S} = (\mathcal{U}, \mathcal{V}, \mathcal{R})$, an *atomic formula* is a formula of the form $X = x$, for $X \in \mathcal{V}$ and $x \in \mathcal{R}(X)$. A *causal formula (over $\mathcal{S}$)* is one of the form $[Y_1 \leftarrow y_1, \ldots, Y_k \leftarrow y_k]\phi$, where

- $\phi$ is a Boolean combination of atomic formulas,
- $Y_1, \ldots, Y_k$ are distinct variables in $\mathcal{V}$, and
- $y_i \in \mathcal{R}(Y_i)$ for each $1 \le i \le k$.

Such a formula is abbreviated as $[\vec{Y} \leftarrow \vec{y}]\phi$. The special case where $k = 0$ is abbreviated as $\phi$. Intuitively, $[Y_1 \leftarrow y_1, \ldots, Y_k \leftarrow y_k]\phi$ says that $\phi$ would hold if $Y_i$ were set to $y_i$, for $i = 1, \ldots, k$.

We call a setting $\vec{u} \in \mathcal{R}(\mathcal{U})$ of values of exogenous variables a *context*. A causal formula $\psi$ is true or false in a *causal setting*, which is a causal model given a context. As usual, we write $(M, \vec{u}) \vDash \psi$ if the causal formula $\psi$ is true in the causal setting $(M, \vec{u})$. The $\vDash$ relation is defined inductively. $(M, \vec{u}) \vDash X = x$ if the variable $X$ has value $x$ in the unique (since we are dealing with recursive models) solution to the equations in $M$ in context $\vec{u}$ (i.e., the unique vector of values that simultaneously satisfies all equations in $M$ with the variables in $\mathcal{U}$ set to $\vec{u}$). The truth of conjunctions and negations is defined in the standard way. Finally, $(M, \vec{u}) \vDash [\vec{Y} \leftarrow \vec{y}]\phi$ if $(M_{\vec{Y} \leftarrow \vec{y}}, \vec{u}) \vDash \phi$.

In addition to the causal setting $(M, \vec{u})$ that describes both the objective causal relations and their actual realization, we also need to represent the agent's beliefs regarding what could possibly happen in order to fill in the **Epistemic Condition**. I do so in the same manner as proposed by HK: we take $\Pr$ to be a probability distribution over a set of causal settings $\mathcal{K}$, so that $\Pr$ expresses the agent's subjective probabilities before the agent performs their action. As do HK, I assume for simplicity that all the causal models appearing in $\mathcal{K}$ have the same signature (i.e., the same exogenous and endogenous variables). We define an *epistemic state* of an agent to consist of a pair $\mathcal{E} = (\Pr, \mathcal{K})$, and define a *responsibility setting* $(M, \vec{u}, \mathcal{E})$ as the combination of a causal setting and an epistemic state.

## 3 The BvH and HK Definitions

BvH [5] work within a game-theoretic framework and do not use causal models, so in order to compare their approach to mine we need to first translate it into the language of causal models. I do not delve into the details but rather offer a rough sketch of such a translation.

Similar to causal models, BvH represent the agents' influence on the outcome $O$ by way of a function. Yet instead of letting some endogenous variables $\vec{A}$ represent the actions of agents directly, they use variables to represent the *strategies* that each agent can adopt to guide their actions. Aside from that, the main difference between the two formalisms is that theirs is unable to represent indirect causal relations. [3] In general, the equations of a causal model allow for an unlimited number of intermediate ancestors between variables $\vec{A}$ and an outcome variable $O$, so that causal influence from an agent's action can be passed on along intermediate variables to the outcome variable. BvH's outcome function on the other hand abstracts away from any mediated form of causal influence, so that the strategies causally determine the outcome *directly*. As a result, their games are to be interpreted as a single-equation causal model of the form $O = f_O(\vec{A})$. (Since the variables $\vec{A}$ are determined directly by the context $\vec{u}$, I adopt the standard practice of leaving their equations implicit.)

BvH use the famous NESS definition of causation that was proposed by Wright [32, 33] – and also formed the inspiration for the Halpern & Pearl (HP) definitions [22, 15, 16] – which states that causes are *Necessary Elements of a Sufficient Set* for the effect. Taking into account the previous remarks, it is more accurate to speak of the *direct NESS* definition. I here present my recent formalization of both the direct and the indirect NESS definitions using causal models [2]. First we need to define causal sufficiency. As do BvH, I take it to mean that a set guarantees the effect regardless of the values of the variables outside of the set.

**Definition 3** (**Sufficiency**). *We say that $\vec{X} = \vec{x}$ is sufficient for $Y = y$ w.r.t. $(M, \vec{u})$ if $Y \notin \vec{X}$ and for all values $\vec{z} \in \mathcal{R}(\vec{Z})$ where $\vec{Z} = \mathcal{V} - (\vec{X} \cup \{Y\})$, it holds that $(M, \vec{u}) \vDash [\vec{X} \leftarrow \vec{x}, \vec{Z} \leftarrow \vec{z}]Y = y$.*

Direct NESS-causation is then defined by stating that:

- the candidate cause and the effect actually occurred;
- the candidate cause is a member of a sufficient set;
- and it is necessary for the set to be sufficient.

**Definition 4** (**Direct NESS**). *$X = x$ directly NESS-causes $Y = y$ w.r.t. $(M, \vec{u})$ if there exists a $\vec{W} = \vec{w}$ so that the following conditions hold:*

DN1. *$(M, \vec{u}) \vDash X = x \land \vec{W} = \vec{w} \land Y = y$.*

DN2. *$X = x \land \vec{W} = \vec{w}$ is sufficient for $Y = y$ w.r.t. $(M, \vec{u})$.*

DN3. *$\vec{W} = \vec{w}$ is not sufficient for $Y = y$ w.r.t. $(M, \vec{u})$.*

We can now formulate the counterpart of the BvH definition using causal models by filling in their conditions into our Responsibility Schema.

**Definition 5** (**BvH Responsibility**). *An agent who performs $A = a$ is responsible for outcome $O = o$ w.r.t. a responsibility setting $(M, \vec{u}, \mathcal{E})$ if:*

- *(**Causal Condition**) $A = a$ directly NESS-causes $O = o$ w.r.t. $(M, \vec{u})$.*
- *(**Epistemic Condition**) There exists $a' \in \mathcal{R}(A)$ so that $\Pr(A = a$ directly NESS-causes $O = o) > \Pr(A = a'$ directly NESS-causes $O = o)$.[4]*

Informally, the BvH definition of responsibility requires that an agent's action directly NESS-caused the outcome, and that the agent believes they failed to minimize the probability of their action causing the outcome. The following example (taken from BvH) illustrates their definition.

**Example 1.** *Two assassins ($Assassin_1$ and $Assassin_2$), in place as snipers, shoot and kill Victim, with each of the bullets fatally piercing Victim's heart at exactly the same moment. Although neither of them could have prevented the outcome, each of them is clearly responsible for Victim's death.*

Let $V$ stand for Victim's death ($V = 1$) or survival ($V = 0$), and let $A_1, A_2$ stand for the actions of the two assassins, where $A_i = 1$ if and only if $Assassin_i$ shoots. We can then capture this example with the single equation $V = A_1 \lor A_2$, and a context $\vec{u}$ such that $A_1 = 1$ and $A_2 = 1$.

---

[3]As I said, this is a rough sketch. Technically, one should distinguish between games in *normal form*, which is the form considered by BvH, and games in *extensive form*, from which the normal form games have been derived. Games in extensive form do allow for indirect relations, and thus there might be a way of representing indirect causal relations in game theory after all.

[4]Of course these probabilities have to be read as being conditioned on the corresponding action, i.e., as "the agent's probability that the action would cause the outcome if it were performed".

Does the BvH definition (Definition 5) succeed in establishing that each of the assassins is responsible for Victim's death? To find out, we first need to evaluate whether $A_1 = 1$ (resp. $A_2 = 1$) directly NESS-causes $V = 1$ (Def. 4). We can choose $\vec{W} = \varnothing$ to get the desired result, as follows.

AC1 is fulfilled because $A_1 = 1$ and $V = 1$ actually happened. AC2 is established by verifying that the following two claims hold: $(M, \vec{u}) \vDash [A_1 \leftarrow 1, A_2 \leftarrow 1]V = 1$ and $(M, \vec{u}) \vDash [A_1 \leftarrow 1, A_2 \leftarrow 0]V = 1$. Since $\vec{W} = \varnothing$, verifying AC3 is easy: we need to find a single intervention on the variables other than $V$ such that they result in $V = 0$. The intervention $[A_1 \leftarrow 0, A_2 \leftarrow 0]$ does the job.

To evaluate the **Epistemic Condition** requires making some assumptions about the assassins's probability attributions. It sounds reasonable to assume that, without evidence to the contrary, each assassin attributed a higher probability to them shooting causing the outcome than them not shooting causing the outcome. Therefore the **Epistemic Condition** is also fulfilled for each assassin, and thus the BvH definition arrives at the right verdict for this example.

We continue with the approach pursued by Halpern Kleiman-Weiner (HK) [17], which uses the modified Halpern & Pearl definition of causation [16]:

**Definition 6** (**HP**). *$\vec{X} = \vec{x}$ HP-causes $Y = y$ w.r.t. $(M, \vec{u})$ if there exists a $\vec{W} = \vec{w}$ so that the following conditions hold:*

AC1. *$(M, \vec{u}) \vDash \vec{X} = \vec{x} \wedge \vec{W} = \vec{w} \wedge Y = y$.*

AC2. *There is a setting $\vec{x}'$ such that $(M, \vec{u}) \vDash [\vec{X} \leftarrow \vec{x}', \vec{W} \leftarrow \vec{w}]Y \neq y$.*

AC3. *$\vec{X}$ is minimal; there is no strict subset $\vec{X}'$ of $\vec{X}$ such that $\vec{X}' = \vec{x}''$ satisfies AC2, where $\vec{x}''$ is the restriction of $\vec{x}$ to the variables in $\vec{X}'$.*

Note that, contrary to the direct NESS definition, the HP definition allows for conjunctive causes $\vec{X} = \vec{x}$, instead of merely atomic causes $X = x$. The minimality condition (AC3) is there to prevent irrelevant events to be added to such conjuncts. We can retrieve a definition of causation for atomic events by simply considering any conjunct $X = x$ that appears in an HP-cause $\vec{X} = \vec{x}$ to be a cause as well, which is indeed what Halpern suggests himself repeatedly [16].

The heart of the HP definition is AC2: it states that the outcome $Y = y$ counterfactually depends on the cause $\vec{X} = \vec{x}$ given that we intervene to hold fixed a suitably chosen set of variables $\vec{W}$ at their actual values $\vec{w}$. To see how this definition works, let us apply it to Example 1.

First we try substituting $\vec{X} = \vec{x}$ with $A_1 = 1$. Alas, this will not allow us to get $A_1 = 1$ as a cause of $V = 1$. We start with choosing $\vec{W} = \varnothing$, and we get that $(M, \vec{u}) \vDash [A_1 \leftarrow 0]V = 1$. The reason is that $\vec{u}$ encodes the actual context, in which $A_2 = 1$, and thus also $V = 1$. Yet what is required for AC2 would be $(M, \vec{u}) \vDash [A_1 \leftarrow 0]V = 0$. The only other choice for $\vec{W} = \vec{w}$ would be $A_2 = 1$, and that does not work either: $(M, \vec{u}) \vDash [A_1 \leftarrow 0, A_2 \leftarrow 1]V = 1$.

Second we try $A_1 = 1 \wedge A_2 = 1$. If this works, then AC3 is satisfied due to the fact that neither of the conjuncts themselves satisfied AC2. $\vec{W}$ has to be $\varnothing$, since there are no other variables. Thus what remains is to find counterfactual values for $A_1$ and $A_2$. As they are binary, the only option is to consider $A_1 = 0 \wedge A_2 = 0$. Clearly, for this choice AC2 is satisfied, as $(M, \vec{u}) \vDash [A_1 \leftarrow 0, A_2 \leftarrow 0]V = 0$. Therefore $A_1 = 1$ is an HP-cause of $V = 1$.

We can now formulate a definition of responsibility that is closely inspired by HK.

**Definition 7** (**HK Responsibility**). *An agent who performs $A = a$ is responsible for outcome $O = o$ w.r.t. a responsibility setting $(M, \vec{u}, \mathcal{E})$ if:*

- **(Causal Condition)** *$A = a$ HP-causes $O = o$ w.r.t. $(M, \vec{u})$.*
- **(Epistemic Condition)** *There exists $a' \in \mathcal{R}(A)$ so that $\Pr(O = o | [A \leftarrow a]) > \Pr(O = o | [A \leftarrow a'])$.*

In addition to disagreeing about the definition of causation, the HK definition also disagrees with the BvH definition about the epistemic condition: rather than requiring that the agent failed to minimize the probability of causing the outcome, the HK definition focuses on the agent failing to minimize the probability of the outcome simpliciter.

Note that both HK and BvH's epistemic condition satisfy our **Responsibility Schema**: an agent who believes that they failed to minimize a probability that they could have minimized, thereby also believes that they could have avoided satisfying the respective epistemic condition. Given that the epistemic condition is a necessary condition for being responsible, they also believe that they could have avoided being responsible for the actual outcome.

Let us apply the HK definition to Example 1. We already established that each $A_i = 1$ is an HP-cause of $V = 1$, so the **Causal Condition** is met. Further, as long as each assassin attributes a strictly positive probability that the other assassin may fail to shoot, we get that $\Pr(V = 1|[A_i \leftarrow 1]) > \Pr(V = 1|[A_i \leftarrow 0])$, so that the **Epistemic Condition** is satisfied as well. (What if the assassins are certain the other assassin will shoot? We come back to this in Section 5.) Therefore the HK definition also arrives at the correct verdict for this example.

# 4 The Causal Condition

Before discussing the problems with NESS- and HP-causation, I present CNESS-causation [2]. As a first step, we define NESS-causation as the transitive closure of direct NESS-causation, which is how it was conceived by Wright [33]. In addition, we pay explicit attention to the path along which the causal influence is transmitted.

**Definition 8** (**NESS**). $X = x$ NESS-causes $Y = y$ along a path $p$ w.r.t. $(M, \vec{u})$ *if the values of the variables in $p$ form a path of direct-NESS causes from $X = x$ to $Y = y$.*

The *Counterfactual* NESS definition (CNESS) takes the NESS definition and adds a subtle counterfactual difference-making condition: there should be a counterfactual value so that it would not NESS-cause the outcome along the same path as the actual value, nor along any subpath.

**Definition 9** (**CNESS**). $X = x$ CNESS-causes $Y = y$ w.r.t. $(M, \vec{u})$ *if $X = x$ NESS-causes $Y = y$ along some path $p$ w.r.t. $(M, \vec{u})$ and there exists a $x' \in \mathcal{R}(X)$ such that $X = x'$ does not NESS-cause $Y = y$ along any subpath $p' \subseteq p$ w.r.t. $(M_{X \leftarrow x'}, \vec{u})$.*

With all the definitions of causation at hand, I now motivate my choice for the CNESS definition by going over some well-chosen examples. We start with a case of Late Preemption.

**Example 2** (**Late Preemption**). *We return to our two assassins, but this time $Assassin_1$ is slightly faster, so that their bullet kills Victim, who collapses and thereby dodges $Assassin_2$'s bullet.*

In this case $Assassin_2$ obviously did not cause Victim's death, and is thus not responsible for the outcome (despite the fact that their act itself is of course still blameworthy). BvH only allow variables for strategies and are thus unable to capture this result, since the asymmetry between both assassins is not a matter of strategy. As illustrated at length by Halpern [16], using causal models this poses no problem. The equation $V = BH_1 \vee BH_2$ expresses the fact that either bullet hitting Victim would be fatal; $BH_1 = A_1$ and $BH_2 = A_2 \wedge \neg BH_1$ captures the asymmetry between both assassins: $Assassin_2$'s bullet only hits Victim if $Assassin_1$'s bullet does not. In the context at hand, we have that $A_1 = A_2 = BH_1 = V = 1$, and $BH_2 = 0$. We now go through the various definitions to verify that $A_1 = 1$ NESS-causes, CNESS-causes, and HP-causes $V = 1$, whereas $A_1 = 1$ does not directly NESS-cause $V = 1$, thereby showing that the direct NESS definition is too simplistic.

We start by verifying that $A_1 = 1$ does not directly NESS-cause $V = 1$. By itself $A_1 = 1$ does not form a sufficient set for $V = 1$, for setting both of the $BH$ variables to 0 guarantees that the Victim survives: $(M, \vec{u}) \vDash [A_1 \leftarrow 1, BH_1 \leftarrow 0, BH_2 \leftarrow 0]V = 0$. In fact, in this context, any sufficient set for $V = 1$ has to contain $BH_1 = 1$, yet $BH_1$ is sufficient for $V = 1$ all by itself. Thus $A_1 = 1$ is not a necessary member of any sufficient set for $V = 1$. Still, $A_1 = 1$ NESS-causes $V = 1$ along $p = \{A_1, BH_1, V\}$, because $A_1 = 1$ directly NESS-causes $BH_1 = 1$ and $BH_1 = 1$ directly NESS-causes $V = 1$.

To establish CNESS-causation requires having a look at the counterfactual setting $(M_{A_1 \leftarrow 0}, \vec{u})$. In this setting we get that $A_1 = 0$, $A_2 = 1$, $BH_1 = 0$, and thus $BH_2 = 1$ (as well as $V = 1$). (Informally: if $Assassin_1$ had not shot, then $Assassin_2$'s bullet would have hit and killed Victim.) Here $A_1 = 0$ directly NESS-causes $BH_1 = 0$, $BH_1 = 0$ directly NESS-causes $BH_2 = 1$ (since it forms a sufficient set together with $A_2 = 1$ and $A_2 = 1$ does not suffice on its own), and $BH_2 = 1$ directly NESS-causes $V = 1$. Therefore $A_1 = 0$ NESS-causes $V_1 = 1$ along $p^* = \{A_1, BH_1, BH_2, V\}$. (Take note of this surprising finding. We come back to it in Example 3.) Since $p^* \nsubseteq p$, we get that $A_1 = 1$ CNESS-causes $V = 1$ (whereas $A_1 = 0$ does not CNESS-cause $V = 1$ in the counterfactual setting, since $p \subseteq p^*$).

To see that $A_1 = 1$ HP-causes $V = 1$, it suffices to note that $(M, \vec{u}) \vDash BH_2 = 0$ and $(M, \vec{u}) \vDash [A_1 \leftarrow 0, BH_2 \leftarrow 0]V = 0$. Lastly, I leave it to the reader to verify that $A_2 = 1$ is not an HP-cause of $V = 1$, and nor is it a direct NESS-cause of anything. Because of the latter, $A_2 = 1$ is not a NESS-cause or a CNESS-cause of anything either.

Modifying the BvH definition so that it uses NESS-causation instead of direct NESS-causation is not a solution, for the NESS definition itself is problematic, as the following example shows. (In the appendix I discuss one more example, a so-called "Frankfurt-case", to show that BvH's reliance on strategies as opposed to events forms another source of problems.)

**Example 3.** *We revisit the counterfactual setting of Example 2 in which $Assassin_1$ does not shoot, so that Victim is killed by $Assassin_2$'s shot.*

We already established for this scenario that $A_1 = 0$ NESS-causes $V = 1$. Thus if we use the NESS definition, we get the absurd result that $Assassin_1$ *failing to shoot causes Victim to die*. If we then supplement the example so that also BvH's **Epistemic Condition** is fulfilled, we get that $Assassin_1$ comes out as being responsible for Victim's death. (Imagine, for instance, that they mistakenly believe to be holding a flare gun that could sound a warning shot so that Victim ducks for cover to avoid $Assassin_2$'s bullet.) We already established that $A_1 = 0$ does not CNESS-cause $V = 1$, the reader may verify that the same holds for the HP-definition.

This leaves CNESS-causation and HP-causation as candidates for the **Causal Condition**. I use Halpern & Pearl's own example to argue against HP-causation [15].

**Example 4** (**Loader**). *"Suppose that a prisoner dies either if A loads B's gun and B shoots, or if C loads and shoots his gun. Taking D to represent the prisoner's death and making the obvious assumptions about the meaning of the variables, we have that $D = 1$ iff $(A = 1 \land B = 1) \lor C = 1$. Suppose that in the actual context $\vec{u}$, A loads B's gun, B does not shoot, but C does load and shoot his gun, so that the prisoner dies. Clearly $C = 1$ is a cause of $D = 1$. We would not want to say that $A = 1$ is a cause of $D = 1$, given that B did not shoot (i.e., given that $B = 0$)." [emphasis added]*

I agree with Halpern and Pearl. A fortiori, $A$ is not responsible for the prisoner's death, even if $A$ only loaded the gun because he was convinced that $B$ would shoot. Now consider the following variant. In the original example, $C$'s shot is determined directly by the context. Imagine we add a little twist, so that $C$ would only fire his gun if $B$ did not, i.e., the equation for $C$ is $C = \neg B$. The above reasoning regarding $A$ still applies, and therefore I believe it is a mistake to all of a sudden consider $A = 1$ a cause of $D = 1$. Yet $A = 1$ now is an HP-cause of $D = 1$ (as it appears in the HP-cause $A = 1 \land B = 0$), and thus $A$ would be considered responsible for the prisoner's death. The CNESS definition avoids this result (as does the NESS definition): the only candidate sufficient set for $D = 1$ of which $A = 1$ could be a necessary part, is $\{A = 1, B = 1\}$. So the mere fact that $B = 0$ in both versions of the example implies that $A = 1$ is not a NESS cause of $D = 1$ in either.

I leave a second counterexample to the HP definition for the appendix and refer the reader to [1] for a detailed critical examination of the HP definition. The alternative definition I there presented is in fact very similar to my CNESS definition, although the precise relation is the subject of further investigation.[5] This leads me to suggest adopting the CNESS definition for the **Causal Condition**.

## 5   The Epistemic Condition

Recall that the difference between HK and BvH's **Epistemic Conditions** lies in whether an action minimizes the probability of *the outcome occurring* (HK) or of *it causing the outcome* (BvH). Given that one cannot cause an outcome unless the outcome actually occurs, and that vice versa, in many cases the best way to make sure that an outcome occurs is by causing it, both of these conditions often go hand in hand. However, as the following example illustrates, they do not always do so, and when they do not the appeal of HK's condition is stronger.

**Example 5. Bombing** *A bomb (B) is connected to three detonators ($D_1$, $D_2$, and $D_3$) by two switches ($S_1$ and $S_2$). $D_1$ is functional if only $S_1$ is on, $D_2$ is functional if only $S_2$ is on, and $D_3$ is functional whenever $S_1$ is on. The equations are thus as follows: $B = D_1 \lor D_2 \lor D_3$, $D_1 = S_1 \land \neg S_2$, $D_2 = S_2 \land \neg S_1$, and $D_3 = S_1$. $Assassin_2$ (reasonably) assigns a probability of 0.6 to $Assassin_1$*

---

[5] I tentatively conjecture that the CNESS definition implies my other definition, and not vice versa.

*turning on $S_1$. He decides to turn on $S_2$, thereby* guaranteeing *that the bomb will explode. $Assassin_1$ decides not to turn on $S_1$, so that the bomb explodes due to the functioning of $D_2$.*

Here we certainly would want to say that $Assassin_2$ is responsible for the explosion, and the reason for this seems to be precisely that he knowingly increased the probability of the bomb going off (from 0.6 if $S_2 = 0$ to 1 now that $S_2 = 1$). There is also no doubt that $Assassin_2$'s action caused the explosion: if he had turned $S_2$ off, the bomb would not have exploded.

However, $Assassin_2$ did act so as to minimize the probability that his act would cause the explosion, regardless of whether one chooses NESS-, HP-, or CNESS-causation. Concretely, for all three definitions of causation, $Assassin_2$'s probability that $S_2 = 1$ would cause $B = 1$ is 0.4, whereas his probability that $S_2 = 0$ would cause $B = 1$ is 0.6. (The details are worked out in the appendix.)

Note that in case $S_1 = 1$, then $S_2 = 0$ would result in the outcome being overdetermined, and thus although the latter action would also be a cause of the outcome, it does nothing to contribute to the probability of the outcome occurring. This is what explains why the two conditions can come apart, and why I take the general moral of this story to be that increasing the probability of the outcome trumps increasing the probability of causing the outcome.

However, it does not follow that the probability of causation is irrelevant, but only that it should fulfill a secondary role. Consider again Example 2, and assume that $Assassin_1$ believes that $Assassin_2$ will shoot, and thus believes that Victim is facing certain death. (If that sounds too unrealistic, imagine $Assassin_1$ is one of ten members of a highly trained firing squad that is executing Victim.) Thus the action of $Assassin_1$ had no effect on the probability of the outcome, and would thus not be responsible for Victim's death according to HK's definition. If $Assassin_2$ has a similar belief, then we end up with nobody being responsible. I take this to be an unacceptable result. (Fischer & Ravizza reach the same conclusion when likewise discussing a case (Missile 2) in which an agent knows that the outcome will ensue no matter what they do, and yet the agent is still responsible for the outcome by choosing to cause it [12, p. 102].)

The lesson I draw from this is that if one knowingly has the opportunity to reduce the probability of causation *without thereby increasing the probability of the outcome*, then an agent is responsible if she fails to do so. Therefore I propose the following definition of moral responsibility.

**Definition 10** (**Responsibility**). *An agent who performs $A = a$ is responsible for $O = o$ w.r.t. a responsibility setting $(M, \vec{u}, \mathcal{E})$ if:*

- (**Causal Condition**) $A = a$ *CNESS-causes $O = o$ w.r.t. $(M, \vec{u})$.*
- (**Epistemic Condition**) *There exists $a' \in \mathcal{R}(A)$ so that one of the following holds:*

    1. $\Pr(O = o|[A \leftarrow a]) > \Pr(O = o|[A \leftarrow a'])$
    2. $\Pr(O = o|[A \leftarrow a]) = \Pr(O = o|[A \leftarrow a'])$ *and*
       $\Pr(A = a$ *CNESS-causes $O = o) > \Pr(A = a'$ CNESS-causes $O = o)$.*

## 6 Degree of Responsibility

My binary definition of responsibility can be complemented with a definition of the *degree of responsibility* in order to capture the widely shared sense that responsibility (as well as blame and praise) is a graded notion. Both BvH's and HK's **Epistemic Conditions** naturally suggest such a definition, and so does my combined condition.

The obvious graded counterpart of HK's condition is to simply look at the *causal effect* [22], which in the context of causal strength is referred to as the *Eells measure of causal strength* of $A = a$ relative to $A = a'$: $CS_e(o, a, a') = \Pr(O = o|[A \leftarrow a]) - \Pr(O = o|[A \leftarrow a'])$ [13, 28]. Sprenger [28] argues for accepting the Eells measure as a general measure of causal strength, which is in line with the priority that my **Epistemic Condition** attributes to it. Moreover, when restricted to positive values, this is in fact HK's definition of the degree of blameworthiness. Likewise, the obvious counterpart of BvH's condition is to look at the increase of probability in causing the outcome. Thus I also define the *actual causation measure of causal strength* as[6]

$$CS_{ac}(o, a, a') = \Pr(A = a \text{ CNESS-causes } O = o|[A \leftarrow a]) - \Pr(A = a' \text{ CNESS-causes } O = o|[A \leftarrow a']).$$

---

[6]Surprisingly, to my knowledge this rather obvious measure of causal strength has been overlooked so far in the literature. (For *any* definition of causation of course, not just CNESS.)

Taking into account that our **Epistemic Condition** is a mixture of those of BvH and HK, I suggest the following definition, where the value of $\alpha$ expresses the relative importance of both measures.

**Definition 11** (**Degree of Responsibility**). *The* degree of responsibility $d$ *for $O = o$ of an agent who performs $A = a$ w.r.t. a responsibility setting $(M, \vec{u}, \mathcal{E})$ is $0$ in case the agent is not responsible, otherwise let $S = \text{argmin}_{a^* \in \mathcal{R}(A)} \Pr(O = o | [A \leftarrow a^*])$, and let $a'' = \text{argmin}_{a' \in S} \Pr(A = a'$ CNESS-causes $O = o | [A \leftarrow a'])$, then $d = CS_e(o, a, a'') + \alpha \cdot max(0, CS_{ac}(o, a, a''))$.*

Informally, this measure works as follows. Among all actions that minimize the probability of the outcome, we take one that minimizes the probability of causing the outcome, and then take a weighted sum of both causal strength measures for that action (where the second measure is ignored if it is negative). This captures the idea that in order to avoid responsibility, the agent should choose an action that makes the outcome as unlikely as possible, and then further select their action so that it makes causing the outcome as unlikely as possible. The following example illustrates this definition.

**Example 6.** *Imagine again our scenario from Example 1, but with the following change: $Assassin_1$ is known to be a reliable assassin, whereas $Assassin_2$ is known to have second doubts and almost never shoots. In other words, it is reasonable for $Assassin_2$ to expect that $Assassin_1$ will shoot, and it is reasonable for $Assassin_1$ to expect that $Assassin_2$ will not shoot. On this particular occasion, both assassins shoot and kill victim.*

Although both assassins are responsible according to my definition, it is easy to see that $Assassin_1$ is responsible to a higher degree: the measures of actual causation are identical for both and so are their respective probabilities of the outcome occurring given that they shoot (namely 1), but $Assassin_1$'s probability of the outcome occurring given that they do not shoot is far lower, and thus[7]

$$CS_e^{Ass_1}(V = 1, A_1 = 1, A_1 = 0) > CS_e^{Ass_2}(V = 1, A_2 = 1, A_2 = 0).$$

Interestingly, recent studies offer empirical confirmation that the agent's epistemic state does indeed impact people's judgments in precisely this way: in a disjunctive scenario (like ours), an agent who performs an action that is *typical* (for them) is considered to be more responsible than an agent who acts *atypically* [18]. The authors contrast this disjunctive scenario, which they have trouble explaining, with a conjunctive one in which both agents' actions are necessary for the outcome to occur, which their account explains quite well. In a conjunctive scenario (in other words, if the equation were $V = A_1 \wedge A_2$), an agent who performs an action that is *atypical* is considered to be more responsible than an agent who acts *typically*, flipping the judgments compared to the disjunctive scenario. That is also the verdict of my degree of blameworthiness: in this case, the atypical agent can reasonably expect the outcome to depend on them performing the action whereas the typical agent can reasonably expect that their action has little impact, which translates into a larger measure of causal strength (both $CS_e$ and $CS_{ac}$) for the former. So in contrast to the account of Kirfel and Lagnado [18], my proposal applies equally to both scenarios and can thus be seen as a formal extension of their work.

## 7 Conclusion and Future Work

Based on a comparison with the work of BvH and HK, I have offered a novel formal definition of moral responsibility that is particularly suited for AI systems by filling in the causal and the epistemic conditions. I used contrasting examples to argue in favor of the Counterfactual NESS definition of causation over the NESS and the HP definition, and in favor of a nuanced epistemic condition that combines the two conditions of BvH and HK. I connected this work to measures of causal strength to define a degree of responsibility. This quantified approach can be further enhanced by also taking into account the robustness of causation, which recent research suggests plays a role in responsibility judgments that is somewhat independent of causal strength [14], as well as by considering the collective responsibility of groups of agents [6, 8]. Lastly, as discussed, a formal definition of responsibility is a necessary prerequisite for definitions of blame and praise. To develop definitions of the latter requires incorporating harm and benefit [3, 4], and possibly also intention. Therefore the current definition can be extended in several ways, which I aim to do in future work.

---

[7]The superscripts $Ass_i$ indicate that we are using each agent's subjective probabilities to assess their degree of responsibility.

## Acknowledgments and Disclosure of Funding

Many thanks to Hein Duijf for helpful comments on an earlier version of this paper, as well as to the Neurips reviewers for their constructive criticism of the original submission. This research was made possible by funding from the Alexander von Humboldt Foundation.

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

## Appendix

### Frankfurt-Case

The following is an example of a so-called "Frankfurt-case", taken from HK. An enormous literature in philosophy is devoted to dealing with these kinds of examples, attempting to reconcile intuitions about responsibility with the counterfactual and causal features that these examples contain. Surprisingly, almost none of it uses causal models, and yet doing so reveals the causal structure to be entirely unproblematic.

**Example 7** (**Frankfurt**). *Imagine Jones poisons Smith, who dies. Unbeknownst to Jones, Black was observing his behavior: if Jones had not poisoned Smith, Black would have given Jones a gun and manipulated him in some way or other so that Jones would shoot Smith. Black is both a perfect observer and manipulator of Jones's behavior, and is thus guaranteed to succeed in his plans. Intuitively it is clear that Jones is responsible for Smith's death, despite the fact that he could not have prevented it. (Typical Frankfurt cases focus on responsibility for an action, as opposed to responsibility for the consequence of an action, and therefore scenarios are normally formulated such that Black manipulates Jones to perform the same action. Except for the shift from the action to the consequence though, those cases are structurally isomorphic.)*[8]

*The* **Epistemic Condition** *of both BvH and HK is obviously fulfilled, for Jones believes that Smith's death is completely dependent on his poisoning. We consider the following equations to assess the causal condition: $SD = JP \lor JS$ to capture the fact that Smith dies ($SD$) if either Jones shoots ($JS$) or poisons ($JP$) him, $JS = BM$ to capture that Jones shoots only when Black hands him a gun and manipulates him ($BM$), and finally $BM = \neg JP$ to capture that Black's action depends on Jones's failure to poison.*

*Regardless of whether we apply the NESS definition, the CNESS definition, or the HP definition, $JP = 1$ comes out as a cause of $SD = 1$, and thus the* **Causal Condition** *is satisfied. (This is easy to see by observing that the structure of this example is a standard case of Early Preemption.)*

*BvH claim that their account can handle Frankfurt-cases like this, but that is a mistake. Recall that their variables represent the agents' strategies rather than their actions, and that we are limited to using a single equation. The outcome function they use when discussing a Frankfurt-case is equivalent to the equation $SD = JP \lor B$, where $B$ represents Black adopting his preferred strategy. Therefore on their account both Jones and Black come out as causes of Smith's death, which is not a sensible result. BvH admit that their NESS definition is unable to handle conditional strategies like that of Black, but contend that since we are here focussing on Jones this is not a problem. Obviously simply stating that one should only focus on the sensible results of one's theory is not a satisfactory way of defending it... (This example also highlights a more philosophical problem with their approach: it is not at all clear what it means for a strategy to be a cause. The broad consensus is that causal relata are either events/omissions or properties of events, whereas conditional strategies are neither.)*

### Counterexample to the HP-definition

We here consider a second counterexample to the HP definition that was suggested in [24]. The example is of particular interest as it was presented precisely within the context of the relation between causation and moral responsibility.

**Example 8.** *We have equations $Y = X \lor D$ and $X = D$, and we consider a context such that $D = 1$. This looks very much like a standard case of overdetermination in which $X = 1$ and $D = 1$ are both overdetermining causes. Yet $X = 1$ is not an HP-cause of $Y = 1$ (and it is a CNESS-cause). The reason for this is that $Y = 1$ depends counterfactually on $D = 1$ by itself, whereas it does not depend on $X = 1$ by itself and nor does it when we take $D = 1$ as our witness $\vec{W} = \vec{w}$. Rosenberg & Glymour [24] argue that this result shows the HP definition cannot offer a basis for moral responsibility, by offering the following scenario to go along with these equations:*

---

[8]This analysis can just as easily be applied to these more typical Frankfurt cases. Still, for those who are sceptical that proponents of Frankfurt cases are equally comfortable as I am with moving from actions to consequences, I point out that Fischer & Ravizza apply this shift in exactly the same manner as I do when discussing responsibility for consequences [12, ch. 4].

*An obedient gang is ordered by its leader to join him in murdering someone, and does so, all of them shooting the victim at the same time, or all of them together pushing the plunger connected to a bomb. The action of any one of the gang would suffice for the victim's death. If responsibility implies causality, whom among them is responsible? ... Halpern's theory says the gang leader and only the gang leader is a cause of the victim's death. This is a morally intolerable result; absent a plausible general principle severing responsibility from causation, any theory that yields such a result should be rejected.*

**Bombing**

We now go through the details for the **Bombing** example. (Ex. 6) We need to consider the following four scenarios:

1. $S_2 = 1$ and $S_1 = 0$
2. $S_2 = 1$ and $S_1 = 1$
3. $S_2 = 0$ and $S_1 = 0$
4. $S_2 = 0$ and $S_1 = 1$

We first go through the details for CNESS-causation.

In scenario 1 we have that $S_1 = D_1 = D_3 = 0$ and $S_2 = D_2 = B = 1$. Here $\{S_2 = 1, S_1 = 0\}$ is sufficient for $D_2$, whereas $\{S_1 = 0\}$ is not. Therefore $S_2 = 1$ directly NESS-causes $D_2 = 1$. Clearly also $D_2 = 1$ directly NESS-causes $B = 1$, and thus $S_2 = 1$ NESS-causes $B = 1$ along $\{S_2, D_2, B\}$. What about the counterfactual setting $(M_{S_2 \leftarrow 0}, \vec{u})$? That corresponds to scenario 3. There, the bomb doesn't even explode (so $B = 0$), and thus there are no causes of $B = 1$. We conclude that in scenario 1 $S_2 = 1$ CNESS-causes $B = 1$.

In scenario 2 we have that $S_1 = S_2 = D_3 = B = 1$ and $D_1 = D_2 = 0$. In this scenario $B = 1$ is directly NESS-caused only by $D_3 = 1$. Since $S_2 = 1$ does not directly NESS-cause $D_3 = 1$, it is not a NESS-cause of $B = 1$.

In scenario 4 we have that $S_1 = D_1 = D_3 = B = 1$ and $S_2 = D_2 = 0$. Here $\{S_2 = 0, S_1 = 1\}$ is sufficient for $D_1$, whereas $\{S_1 = 1\}$ is not. Therefore $S_2 = 0$ directly NESS-causes $D_1 = 1$. Clearly also $D_1 = 1$ directly NESS-causes $B = 1$, and thus $S_2 = 0$ NESS-causes $B = 1$ along $\{S_2, D_1, B\}$. What about the counterfactual setting $(M_{S_2 \leftarrow 1}, \vec{u})$? That corresponds to scenario 2, in which $S_2 = 1$ does not NESS-cause $B = 1$. So $S_2 = 0$ CNESS-causes $B = 1$ in scenario 4.

As a result, if $Assassin_2$ chooses $S_2 = 1$, the probability of CNESS-causing $B = 1$ is the probability that $S_1 = 0$, which is $0.4$. By contrast, if $Assassin_2$ chooses $S_2 = 0$, the probability of CNESS-causing $B = 1$ is the probability that $S_1 = 1$, which is $0.6$.

NESS-causation for each scenario is already discussed in the above, so we move on to consider HP-causation. In scenario 1 we have counterfactual dependence of $B = 1$ on $S_2 = 1$, and it is well-known that this suffices for HP-causation (as well as for CNESS-causation, by the way [2]).

In scenario 2, note that $D_3$ suffices for $B = 1$, and thus satisfying AC2 is possible only when either $D_3 = 1$ or $S_1 = 1$ is also part of the candidate cause $\vec{X} = \vec{x}$. However, $B = 1$ counterfactually depends on $D_3 = 1$, meaning that $D_3 = 1$ is a cause all by itself. Thus $\{S_2 = 1, D_3 = 1\}$ is not minimal, and because of AC3 this means that it is not a cause. That leaves $\{S_2 = 1, S_1 = 1\}$. But this is not minimal either, for $S_1 = 1$ is a cause all by itself: one can take $\vec{W} = \{D_2\}$ as a witness to get $B = 0$ when $S_1$ is set to 0. Therefore $S_2 = 1$ is not part of any cause of $B = 1$.

Since $B = 0$ in scenario 3, $S_2 = 0$ does not HP-cause $B = 1$ there either, leaving scenario 4. As with scenario 2, the candidate cause will have to include $D_3 = 1$ or $S_1 = 1$. Contrary to scenario 2 though, $D_3 = 1$ is no longer a cause by itself, since $D_1 = 1$ holds, and will remain to hold also when we set $D_3$ to 0. Since $B = 1$ counterfactually depends on $\{S_2 = 0, D_3 = 1\}$, we get that each of them HP-causes $B = 1$.

