# OpenReview forum: "Moral Responsibility for AI Systems"
_NeurIPS.cc/2023/Conference — NeurIPS 2023 poster_

### Official Review · Reviewer_ZjtP · 2023-06-30

**Soundness:** 3 good
**Presentation:** 3 good
**Contribution:** 3 good
**Rating:** 7
**Confidence:** 1

**Summary:**

In this paper the authors put forward a formalised definition of moral responsibility that can be applied to AI systems. To do so, they compare and contrast how existing definitions fare with regard to a causal and an epistemic condition for responsibility. Using these contrasting cases, the authors argue for a form of counterfactual ‘Necessary Element of a Sufficient Set’ definition of causation, and combine two existing definitions of the epistemic condition.

**Strengths:**

I should note that I am less able to assess the technical aspects of the paper, and so I focus on the conceptual aspects.

Originality: The paper advances a novel formalisation of responsibility, which is contrasted clearly and well with existing definitions of the causal and epistemic conditions for responsibility.
Quality: As I’m not less able to assess the technical aspects of the paper, I do not believe that I am well-placed to comment on the technical quality. Conceptually, the paper is strong.
Clarity: The paper is well-written. (Very minor, but after each example large sections of text are left in italics. This makes it difficult to read and to distinguish between the example and the discussion of the example.)
Significance: Both the causal and epistemic conditions for responsibility are debated in the wider literature, where a challenge is to capture important nuances in formalisations. If successful, the contribution is significant.

**Weaknesses:**

I do not feel competent to comment on the technical substance of the paper.


**Questions:**

My questions are general conceptual questions, not ones that necessarily need to be addressed.

At the end, the authors indicate that the quantified approach taken could be applied to collective responsibility. How do the authors anticipate the approach dealing with the epistemic condition regarding collective responsibility, in particular? What about institutional agents and responsibility?

Blame and praise are inherently most value-infused than a causal notion of responsibility. How would the authors proceed to capture these aspects, and is it necessary?

---

> ### Author Rebuttal · Authors · 2023-08-05
>
> We thank the reviewer for their positive and encouraging feedback!
>
> The reviewer raises a good point that the epistemic condition in particular poses some challenges once we generalize to a notion of collective responsibility. This issue is not novel though, as research in social epistemology (and in social choice theory) also focusses on collective epistemic states, so in first instance we could consult this literature for inspiration. One option would be to integrate communication between the agents into our framework, which definitely seems realistic within the context of AI systems (just think of autonomous vehicles, or swarms of drones, for example). Thus, the various agents can coordinate their actions, and in this sense be treated more or less as if they form a single agent, somewhat similar to how the law treats corporations. But these are merely initial thoughts on the subject, we are definitely open to suggestions on this topic.
>
> Regarding blame and praise: as we mention in our paper, we take our definition to at least form a necessary condition for blame and praise. We offer one line of thought on how to construct a definition of blame in our reply to reviewer ir7c, when discussing how the recent work "A Causal Analysis of Harm" could be combined with our approach to form a definition of blame. (We add a bit more detail in our reply to reviewer ir7c.) A natural further suggestion would be to do the same for the notion of benefit, giving a definition of praise. We would need to integrate further conditions than merely harm and benefit though, because in and of themselves those do not deal with the issue of how to compare and weigh off multiple outcomes (think of trolley cases, the doctrine of double effect, etc.), or on how to integrate the "cost" that an agent incurs by performing an action, nor does it fully capture what duties and obligations an agent might have. In other words, generalizing to blame and praise would be an exciting but far from trivial avenue to pursue.
>
> Lastly, we apologize for the long sections of italics, as we now realize they are not very readable, we will correct this in the final version.

---

### Official Review · Reviewer_UYHL · 2023-07-03

**Soundness:** 4 excellent
**Presentation:** 3 good
**Contribution:** 2 fair
**Rating:** 6
**Confidence:** 3

**Summary:**

This paper formalizes the notion of responsibility of an agent by using
contrastive necessary element of a sufficient set causation (CNESS) for the
causal condition and Halpern and Kleiman-Weiner's notion epistemic
responsibility.  The resulting formulation of responsibility benefits from the
strong alignment of CNESS with intuitions of causation/responsibility vis-a-vis
simple NESS and Halpern and Pearl-causation.  The paper extends responsibility
from a binary variable to real-valued degree of responsibility.


**Strengths:**

## Originality
## Quality
- `[major]` Incorporates the strengths of multiple components to formulate an effective formalization of responsibility.
- `[minor]` Formulation allows for degrees of responsibility.
## Clarity
- `[minor]` Examples illustrate the differences between theories of causation and epistemic responsibility.
## Significance


**Weaknesses:**

Although the formulation of moral responsibility is sound on philosophical
grounds, the paper does not make a strong case for how this applies to AI.
Thus, this paper seems out-of-scope for NeurIPS.


**Questions:**

How is formulation of responsibility particularly well-suited for AI?  What
impacts could it hav?

## Comments
- `Example 2` The paragraphs of italicized text are a bit distracting.


**Limitations:**

If arguing for a definition of moral responsibility, it is important to at
least briefly touch on the societal impacts of the work vis-a-vis the
contributions.

---

> ### Author Rebuttal · Authors · 2023-08-05
>
> We must say that we were very surprised to read this review. The reviewer recommends a strong rejection, and yet they fail to point out a single flaw in our paper. (For completeness, we should point out that their summary does contain a flaw, as we do not use HK's epistemic condition but rather present a novel condition of our own.) Moreover, the entire basis for their verdict lies in their comment that they fail to see how responsibility is relevant to AI, and thus they judge our paper out of scope for NeurIPS.
>
> Is the reviewer not convinced that the ethics of AI is a crucially important topic to do research on, and thus deserves a place at the most important conference on AI? Because if so, then we must firmly disagree (and as reviewer 7Pui outlines in their review, there is a wealth of literature within AI to back this up): we believe that addressing the ethical challenges posed by the use of AI are of vital importance and require much more rather than less research, given that the stakes could not be higher. Moreover, we are confident that this is also the predominant view nowadays within the AI community, and we are therefore at pains to understand the reviewer's succinct and dismissive judgment of our paper.
>
> We develop a notion of responsibility that is formulated using nothing but causal models and probabilities, both of which are well-established formalisms within AI, and one that only requires a very minimal notion of agency to be applicable, thus making it well-suited for the kinds of artificial agency that one could attribute to even rather limited AI systems. We see this as well within the scope of NeurIPS, and we therefore kindly ask the reviewer to explain why they believe otherwise.

---

> > ### Comment · Reviewer_UYHL · 2023-08-11
> > **Ethics is important but paper does not apply it to AI**
> >
> > I have read the author rebuttal.
> >
> > > The reviewer recommends a strong rejection, and yet they fail to point out a single flaw in our paper. Moreover, the entire basis for their verdict lies in their comment that they fail to see how responsibility is relevant to AI, and thus they judge our paper out of scope for NeurIPS.
> >
> > I did not find any major flaws in the paper _per se_, and I think that the paper makes good contributions which are well supported---in the context of philosophy.
> > I do not "fail to see how responsibility is relevant to AI", but I fail to see how this paper contributes to the bridge between the philosophy of responsibility and AI.
> > A contribution along these lines would do more to show how the proposed formulations of responsibility tangibly connect to the practice of AI.
> > For example, one might argue how allowing for degrees of responsibility, autonomy, etc. are necessary for applying to AI since they present a greater range and variety of capability vis-a-vis humans.
> > Or another direction might be showing how the formulation of responsibility can be directly plugged into current ML paradigms.
> > Either way, a practitioner of deep learning should be able to walk away from this paper with a least some sense of how this notion of responsibility is to be applied to the practice deep learning.
> > So far as I can tell, the only steps that this paper takes in this direction is mentioning in the introduction that AI can fulfill the preconditions for responsibility.
> > My judgment, then, is that this paper would have to do significantly more "bridging" between its solid formulation of responsibility and the practice of AI to be relevant to NeurIPS.
> >
> >
> > > Is the reviewer not convinced that the ethics of AI is a crucially important topic to do research on, and thus deserves a place at the most important conference on AI?
> >
> > I am convinced that this is crucially important, but the "of AI" part of this ethics paper does not feature strongly enough.
> >
> > > We develop a notion of responsibility that is formulated using nothing but causal models and probabilities, both of which are well-established formalisms within AI, and one that only requires a very minimal notion of agency to be applicable, thus making it well-suited for the kinds of artificial agency that one could attribute to even rather limited AI systems.
> >
> > If about 1/4 to 1/3 of the paper were spent discussing what is mentioned above (instead of just in the beginning of the introduction and conclusion), I think this paper would easily be a 7/10 or 8/10.
> > Since space is, of course, limited, some of the more weedy (but not unnecessary) philosophical issues could be bumped to the appendix---the main arguments would be intact for the general audience, and for the philosophically inclined, they could read on in the appendix.

---

> > > ### Author Response · Authors · 2023-08-12
> > >
> > > We thank the reviewer for clarifying their reasoning, this is very helpful!
> > >
> > > We agree that we could say more on the explicit connection to AI, and we are definitely willing to do so. (Note that we have an extra page in the camera-ready version, which helps with the space issue.) The instructions for the discussion period were to be brief in our responses, and therefore we here only mention at a very high level some preliminary thoughts on this. However, if the reviewer would like us to elaborate, we can certainly do so.
> > >
> > > 1: A good formal definition of responsibility _simpliciter_ is in and of itself highly relevant to having a good definition of responsibility for AI systems, since we would like the latter to be closely informed by the former.
> > >
> > > 2: By minimizing the role of the Control Condition, our definition requires only a very weak sense of agency, making it particularly suitable for AI systems. (In particular, we can connect our definition to the more pragmatic side of the debate on Artificial Agency.)
> > >
> > > 3: Causal models and probability are well-known to AI researchers, in contrast to many more philosophical and informal notions that are used in traditional definitions of responsibility, thus  our definition can be more easily understood and applied by AI researchers than other definitions.
> > >
> > > 4: Our definition can easily be integrated into AI systems that are _already_ explicitly making decisions informed by a causal model. This allows for an AI system to itself reason about responsibility when making decisions. (Also, as mentioned in our reply to reviewer ir7c, it can be integrated together with "A Causal Analysis of Harm" (NeurIPS 2022), and this could be the starting point of a  toolbox for ethical reasoning. Furthermore, they have a paper on "Quantifying Harm" (IJCAI 2023) and this sits well with our degree of responsibility, because -- as the reviewer points out -- degrees matter when comparing the kind of fine-grained decisions that an AI is capable of performing and reasoning about.)
> > >
> > > 5: More generally, a "regulatory" AI system that has a causal model of the salient moral factors in some domain of application can use our definition to evaluate _other_ AI systems that are operating within that domain, by first querying that system in an efficient way so as to extract a representation of the epistemic state that is informing the system (which is not trivial of course!), and then evaluate its decisions by using these two components and applying our definition.
> > >
> > > We can offer more details on each of these issues if the reviewer so desires.

---

> > > > ### Comment · Reviewer_UYHL · 2023-08-17
> > > > **Proposed changes can make contributions more relevant to AI practitioners**
> > > >
> > > > Although I am not well-versed on the direct application of causal models to AI, I am generally satisfied with the proposed changes in the above comment as I think they make explicit the ways in which the contributions of the paper are to be interpreted by members of the AI community.  While I think an integration of these points from the beginning the paper would have yielded the best results, these additions are sufficient for me to recommend acceptance, although only tentatively since I will not be able to see the final revision of the paper with these changes.  I am changing my rating from a 2/10 to 6/10 and keeping my confidence at a 3/5.

---

### Official Review · Reviewer_ir7c · 2023-07-04

**Soundness:** 3 good
**Presentation:** 3 good
**Contribution:** 3 good
**Rating:** 7
**Confidence:** 2

**Summary:**

This paper presents a novel formal definition of moral responsibility tailored for AI systems, filling in both causal and epistemic conditions. The work effectively draws comparisons to BvH and HK's works, favoring the Counterfactual NESS definition of causation and a nuanced epistemic condition.

**Strengths:**

 The proposed formal definition of moral responsibility for AI systems is a novel contribution. The paper contrasts several existing theories of causation and epistemic conditions to make its case, demonstrating a thorough understanding of the existing literature. The submission is clearly written and well organized. The authors articulate complex concepts with clarity and a lot of interesting examples.

**Weaknesses:**

There is a need for further elaboration on how this proposed concept can be practically applied and measured in real-world AI systems. Also, the work would be more convincing if it incorporated a broader range of philosophical perspectives on moral responsibility. Lastly, while the paper talks about future work related to defining blame and praise, these aspects are not explored in the current submission, leaving an incomplete picture of the potential implications of their framework.

**Questions:**

I would like to see more discussions on the application of the proposed definition in real-world scenarios.

**Limitations:**

 The authors have acknowledged that future work could enhance the quantified approach by considering collective responsibility and degree of causation. However, practical application and testing of the proposed definition are yet to be performed.

---

> ### Author Rebuttal · Authors · 2023-08-04
>
> We thank the reviewer for their comments, and their positive evaluation of our paper!
>
> Regarding their question about real-world scenarios: we actually completely agree! We would also like to see more discussion of our definition to real-world scenarios, and we very much would like to offer this in future work. In particular, there is now a very large empirical literature in experimental philosophy and in psychology on people's actual judgments of responsibility and causation in many real-life scenarios. Although there is an abundance of empirical results, there is much disagreement on their interpretation. Concretely, there is disagreement on whether -- and how -- causal judgments can be separated from judgments of responsibility, on what the role is of the epistemic states of the agent (see in particular the recent work by Kirfel and Lagnado that we discuss in the supplementary material), what the role is of  probabilities and background conditions, and on what the right view is on actual causation. In the supplementary material we only briefly sketch how our work could shed some new light on all of this. We would very much like to write a follow-up paper in which we apply our definition to this vast literature, because we are optimistic that it could offer a more principled interpretation than what is currently on offer.
>
> Of course there also exist other kinds of real-world scenarios that we should look into, namely the kind of scenarios in which an AI system might realistically be deployed to make the kind of decisions to which our definition would apply. Autonomous vehicles and the oft-discussed trolley cases come to mind (although these would require extending our definition to include a focus on multiple outcomes), as well as the use of autonomous weapons and the moral requirements that are implied by principles of just war, to name just a few.
>
> More generally, recently Beckers, Chockler, and Halpern have developed "A Causal Analysis of Harm" (NeurIPS, 2022), which could be integrated into our framework as well (except that we would choose to replace their definition of causation with the CNESS definition). At first thought, one might speculate that an agent who performs action A is blameworthy for some outcome O  whenever:
> - the agent in question is responsible for O in virtue of performing A (which is the focus of our analysis), and
> - A caused harm to another agent, where the harm caused is due to A causing O (which is the focus of Beckers, Chockler, and Halpern).
>
> Lastly, as we have outlined in our response to reviewer 7Pui, we completely agree that our definition focuses only on one kind of responsibility and should be placed within a broader landscape that discusses important related (but distinct) notions such as blame, accountability, liability, and others. We will be more explicit in the paper regarding the relation of our work to this broader landscape.

---

### Official Review · Reviewer_7Pui · 2023-07-17

**Soundness:** 2 fair
**Presentation:** 2 fair
**Contribution:** 2 fair
**Rating:** 3
**Confidence:** 3

**Summary:**

The paper aims to link causal and moral responsibility through formal modeling using structural causal models. Engaging with existing literature on causality, the paper argues for the benefits of certain definitions over others in introducing a 3-part _responsibility schema_ based on a control condition (that an agent had control over the causal action), a causal condition (that the controlled action caused the outcome), and an epistemic condition (the agent believed that their control choice minimized the probability of an alternative, less preferred outcome, thereby avoiding responsibility for the actual outcome). In support of this model, the paper shows several example scenarios that differentiate existing definitions that could be used to operationalize this schema.

**Strengths:**

+ The linking of causal and moral responsibility is useful and important.
+ Connecting reasoning about values to formal models is difficult but a promising research direction.
+ The paper engages deeply with existing definitions of causality and its relationship to agents' beliefs.
+ The paper is well organized in support of its contribution, the responsibility schema.

**Weaknesses:**

- The paper's view of moral responsibility is not presented as substantially distinct from causal responsibility and the connection to human values or moral philosophy is extremely weak. This is disappointing given that there is a large literature on linking causal responsibility to the value of accountability _within the context of AI systems_ that has been developing for many decades and now constitutes one of the main lines of thought in two major venues (ACM FAccT and ACM/AAAI AIES), which are wholly ignored in the discussion and citations (see, e.g., various works on accountability of, e.g., Kroll and also various works on liability of, e.g., Selbst). I think this context should be acknowledged, at a minimum. If it's not relevant, that's acceptable but the paper should say why and mention it as an alternative approach.
- Although the paper aims to link causal and moral responsibility, it ignores several classical lines of thought attacking this question. For example, a classical question in law going back hundreds of years is the distinction between causal or moral responsibility and _liability_, the question of which agent the law should assign responsibility to (a theory in which many long-discussed cases directly contravene examples in the paper). Relating these ideas to computing systems, nearly 30 years ago, Nissenbaum's classic "Accountability in a Computerized Society" addressed the question of when causal and moral responsibility could be separated and why they often can't be _in computing systems specifically_. There is even a recent re-establishment of this argument by Cooper, Moss, Laufer, and Nissenbaum revisiting the framework in light of data-derived decision rules. Again, the fact of an alternative approach should inform the location in the literature of the paper's contribution. What is the relationship between the proffered responsibility schema and values such as accountability, regimes such as liability, or concepts such as moral responsibility distinct from causal responsibility?
- I think the paper, by sticking too closely to the details of definitions in the literatures of formal causal calculi and structural causal modeling, falls victim to some of the standard criticisms of this literature and specifically of tying its results to reasoning about human values. If the paper wishes to claim such a tie as a contribution, this context must also be reckoned with. For example, authors such as Hu, Kohler-Hausmann, Miller, and Kasirzadeh and Smart have in the last few years attempted to build theories that deal with multiple-causation and "structural causation" (here, structural in the sense that the cause is inherent to the shape of an entire system, rather than in the sense that the structure of the graph model informs which variables are or are not causally relevant). Such approaches resist reducing causation to even a set of variables or to sufficient conditions, and have proven useful in understanding things which are otherwise difficult, such as describing the plausibility of different counterfactuals. Again, there is an alternative approach in the literature and the contribution would be strengthened by acknowledging it and setting off the contribution in light of existing work. Not everything tangentially relevant has to be cited, of course, but the paper should make clear where its contributions lie and why other work with similar goals is different.


**Questions:**

* Where do the authors see their contribution and how do they set off alternative approaches in other literatures? What of other concepts such as accountability, liability, or answerability for outcomes?
* Is moral responsibility equivalent to causal responsibility in a modeled context? Why or why not?
* How does the offered model manage situations where causes aren't instrumental but rather structural and systemic? What of multiple-causation, both in general and in these contexts?

**Limitations:**

The work should overall be better contextualized around the full richness of the problem used to motivate it.

---

> ### Author Rebuttal · Authors · 2023-08-04
>
> We thank the reviewer for their detailed engagement with our paper, but we fail to understand why they recommend rejection. The core contribution of our paper is an analysis that is an improvement over two important existing analyses. As they do not point to any concrete flaws in our analysis, it appears the reviewer’s judgment is based mostly on the absence of a more detailed integration into the broader literature. Although we are unsure how we could do justice to such an integration in a short conference paper, we will add more context and references to better situate our contribution.
>
> Questions:
>
> Q1:
> The reviewer is completely right that there is more to responsibility than what is discussed in our paper. Moreover, they are also right in pointing out the importance of many related concepts. The aim of our paper is therefore fairly modest, in the sense that it focusses _only_ on responsibility for an outcome in virtue of performing an action. Furthermore, we point out that our notion of responsibility in and of itself leaves open entirely whether the agent is thereby blameworthy or praiseworthy: instead the definition here presented is a necessary condition for such further judgments, but certainly not a sufficient one. Therefore our definition is in no way meant to offer an _exhaustive_ theory of all kinds of responsibility for AI systems, let alone a theory of accountability and other related notions. Rather, we see our work as filling in just one element of such a theory, which we believe can be integrated into many existing theories out there. (See below for more details.) We will be more explicit on this point.
>
> Q2:
> If causal responsibility means “being an actual cause”, then in our view — and for the notion of responsibility that we are focussing on! — such causal responsibility is a necessary condition for moral responsibility, for it is captured by our Causal Condition. We take up this view from the philosophy literature, and it is also well-supported by the law. Given that, on our view, moral responsibility also requires meeting the other two conditions, it is therefore not equivalent to causal responsibility.
>
> Q3:
> Our definition exclusively focuses on actual (aka token) causation, and exclusively on variables which represent the actions of an agent. Thus, at first sight, any responsibility that is owed due to there being structural or systemic causes (which are, we take it, not actual, and are not actions), is not covered by our definition. If the reviewer has in mind a particular kind of example where this seems problematic, we would be very much open to discussing it.
> Regarding multiple causation: since the only causes that are relevant to our definition are the actions of agents, issues of multiple causation only show up in case of multiple actions. Our Assassins example in which both shoot can be seen as an instantiation of multiple causation. In the current paper we only judge their responsibility individually.
>
> Remarks:
>
> We do not see the connection to moral philosophy as being very weak, given that we are starting out from a responsibility schema that is derived directly from the standard literature in moral philosophy. We therefore disagree with the reviewer that this schema captures the essence of our contribution: our contribution lies in _filling in_ this schema in a manner that we argue is superior to existing accounts. Furthermore, we do not consider this to be an _alternative_ to the important work that the reviewer mentions: it is merely one piece of the full picture that should inform the moral evaluation of AI systems.
>
> We are unclear as to what the reviewer means when they state that there exist classical lines of attack on the view that moral responsibility is closely connected to causation. (To reiterate, we make no claim to exhaustivity: we acknowledge that there exist notions of responsibility that are not so closely tied to causation.) They refer to work on liability, which is not the topic of our paper, and they refer to work by Cooper et al that seems to support the existence of a close connection. In fact, we looked into the article mentioned, and we believe our work could be very beneficial for the issues there identified. For example, they state that "Blame, defined in terms of causation and faultiness, is assigned to moral agents for harms they have caused due to faulty actions", which is entirely in line with our view (see also our reply to reviewer ir7c regarding harm). Later they say that "in practice, if a trained model causes harms, it can be extremely challenging to tease out particular actors who should answer for them." Our work is meant precisely to tease out those actors, and thus we see it as fitting perfectly within the challenges identified by Cooper et al. The reviewer also points to the work by Kroll, and there too we see a connection, as he explicitly focusses both on causal responsibility as well as on the agent’s ability to perform alternative, preferable, actions, as offering one important way to understand accountability. So we fail to see how our work is in conflict with the literature on accountability that the reviewer is referring to.
>
> Lastly, our reliance on causal models is a consequence of their widespread use in the study of causation, both in AI as well as in philosophy. Despite their limitations, causal models are still the most popular (and successful, we might add) formalism for addressing causal issues. We would like to learn more as to which criticisms the reviewer has in mind in particular, because as far as we know, these criticisms mainly target counterfactuals and causation that involve causally suspicious variables such as race or gender, and do so in causal models that partly aim to capture the causal structure of society. Our approach does no such thing: we focus on mundane cases of actual causation (rather than structural or type causation), in which the actions of agents cause specific outcomes.

---

> > ### Comment · Reviewer_7Pui · 2023-08-21
> >
> > I should say, I firmly disagree with the notion that a review must identify deficiencies within the four corners of the paper. When work is not sufficiently connected to surrounding literature or scopes claims incorrectly based on available ideas, that is a holistic assessment and requires seeing the work in context. I'm not asking that this be a survey, but rather that the contribution be contextualized and that, where the contribution differs from existing ideas, that this gap be justified.
> >
> > I agree that engaging the causal modeling literature is valuable for the reasons the authors mention: it's the method that gets used in AI research and maybe also practice. I'm a bit confused by the authors' claims that the surrounding literature pointed out in the initial review remains irrelevant - engagement with this work in the comment _demonstrates_ its relevance. By "alternative", I meant less that these works provide distinct frameworks and more that other methods (e.g., critical scholarship/critical theory) are yielding related ideas and it is important to fit the ideas in this paper into the broader context, whether the related work comes to the same conclusions or not. I recognize that there are a lot of contexts the work could be placed within. My argument is that some deference needs to be given to all of them, but perhaps not to the same depth. So for example, it would be reasonable to set aside legal scholarship on liability, which is a huge area that probably isn't super relevant. Work applying ideas about causal and moral responsibility _to AI_ is more relevant, and should be relied on more heavily (this is the essence of the claim in the initial review). This is especially true given that the claim that the work can and should be much more closely tied together is part of the response to another review (UYHL). On the other hand, the work is deeply contextualized in the causal modeling literature, which I think is fantastic, although I'm not well equipped to evaluate the quality of this.
> >
> > The comment suggests heavily that the related work comes to the same conclusion as this paper, but my point in the initial review was the opposite: although the comment claims otherwise, existing work *separates* causal and moral responsibility. The function of the example of liability - which I concur is not the focus of this work and probably shouldn't be - was to illustrate how different facets of accountability can be separated in practice and often are. Work on liability is relevant insofar as cases where different actors are causally vs. morally responsible vs. liable have been considered under the banner of liability and this is ignored in the work, which claims that causal responsibility is an antecedent condition for moral responsibility (if so, how could they be separated? How could an entity be liable without being causally or morally responsible, and if that's a mistake in the analysis what is wrong with existing work on liability?). This is what I mean by the connection to moral philosophy being weak - it's central to much work that these separations are possible, so rejecting them is a heavy lift, one I don't think is made by this paper.
> >
> > I do think the claims are overbroad, partly for the reasons just above and even more because, when I challenged the applicability of the blanket claim to scenarios of structural causation, the authors responded that "our definition exclusively focuses on actual (token) causation". This is in conflict with the claim at 66 that the responsibility schema applies to "all definitions of responsibility that we aim to consider" (a quick review of the intro does not give a clear bound to the restriction at the end of the sentence, leaving me to conclude that the schema is meant to apply broadly). I still think the claims need to be narrowed, and in the service of contextualization, such narrowing could take the form of an argument that the schema applies only to situations where actual causation applies and other work demonstrates that this is not all or even many of the situations in which accountability questions apply to AI. So the claim should be narrowed somehow, consistent with the paper's "modest" aims.
> >
> > I observe a gap between my claim in the review (that issues of structural/systemic cause are under-attended in the paper) and the response (that these are essentially asking the paper to consider type causation, which is distinct but perhaps also important?). The persistence of this gap furthers my view that the context of relating responsibility issues to AI is quite weak in the submitted version.
> >
> > Indeed, the extent to which the application of the schema _to AI_ is critical to the contribution of the paper. I still believe that the project of "_filling in_ the schema" has to be justified in terms of how well the filling in applies to the specifics of the scenarios at issue (i.e., the structure of AI systems). Without engaging the context, it's hard to evaluate this.

---

> > > ### Author Response · Authors · 2023-08-21
> > >
> > > We thank the reviewer for their continued engagement with our paper.
> > >
> > > "Identify": We never claimed that a review should identify deficiencies within the four corners of the paper, but merely that the absence of a single concrete deficiency is hard to square with recommending a rejection. In their response the reviewer still did not identify any deficiency with our analysis, instead the criticism is directed at the lack of contextualization with regards to related work. Therefore we would like to once again emphasize, as we did in our original rebuttal (and also in our rebuttal to reviewer UYHL), that we are perfectly willing to add more background context to better situate our definition within the literature and to be explicit about the scope of the notion of responsibility that we are focussing on.
> > >
> > > "Relevance": We are unsure what the reviewer is referring to, as the word “irrelevant” does not appear in our rebuttal. With respect to the work that is critical of using socially constructed variables or ignoring structural causation, we merely meant to make clear that as far as we can tell, that criticism in no way applies to the kinds of variables and causal claims that we make use of in the paper, which is why we asked the reviewer to learn more about the specific criticism that they have in mind.
> > >
> > > "deference": We entirely agree. As we tried to make clear in our rebuttal, we do not see our approach as being in conflict with the work mentioned in the review, but rather as offering one part of the bigger picture. Moreover, we tried to indicate how our view on responsibility fits quite naturally within two strands of work that the reviewer mentioned, that by Cooper et al., and that by Kroll.
> > >
> > > "Separate": We likewise separate moral and causal responsibility, since causation is but one condition of our definition. As we mentioned in our rebuttal and illustrated through various citations, we failed to find arguments in the articles mentioned for the complete separation of causal and moral responsibility, but rather found support for the view that causation is an important component of responsibility. Moreover, when it comes to the philosophical literature on _moral responsibility for outcomes_, which is the focus of our work, causation is almost universally taken to be a necessary condition. (See our citations [5,18,13,3]. The only exception we are aware of is Sartorio, and her disagreement stems entirely from her disagreement with the causal verdicts reached in certain examples.) Concretely, this is a quote from the Stanford Encyclopedia of Philosophy entry on Moral Responsibility, which clearly implies that causation is taken to be a necessary but insufficient condition for the responsibility for outcomes: “Moral responsibility should also be distinguished from causal responsibility. … the powers and capacities that are required for moral responsibility are not identical with an agent’s causal powers, so we cannot infer moral responsibility from an assignment of causal responsibility. … morally responsible agents may explain or defend their behavior in ways that call into question their moral responsibility for outcomes for which they are causally responsible. Suppose that S causes an explosion by flipping a switch: the fact that S had no reason to expect such a consequence from flipping the switch might call into question his moral responsibility ... for the explosion without altering his causal contribution to it.’’
> > >
> > > "Liability": As to the separation of liability and causal responsibility, all we can do is reiterate that our focus is on the moral responsibility for outcomes, and that we acknowledge the existence of different conceptions of responsibility. Responsibility is a single term that captures several concepts, and we do not disagree that some forms of liability (such as that which arises due to contractual agreements) do not require a causal connection between the responsible agent and some outcome.
> > >
> > > "Actual causation": We fail to see the reviewer’s point: the causal condition in our schema involves particular events, which is per definition what actual causation is concerned with. All three of the NESS, the CNESS, and the HP definition, are examples of definitions of actual causation.
> > >
> > > "Narrowed": As we tried to indicate in our rebuttal, we are definitely prepared to add more context to our paper so as to better delineate its scope, and be more explicit about the fact that  the notion of responsibility is broader than moral responsibility for particular outcomes.
> > >
> > > "Gap": We are unaware of work that discusses responsibility due to type causation, and we would be curious to hear more about this. As outlined in our reply to reviewer UYHL, we disagree that the connection to AI is weak. An AI system making particular decisions or performing particular actions that cause certain outcomes is a widely prevalent phenomenon, and it is to such scenarios that our definition applies.

---

> > > > ### Comment · Reviewer_7Pui · 2023-08-22
> > > >
> > > > Rather than lawyer through past language, I'm going to focus on the text of the paper that was submitted following the confusion indicated by the comments. Also, it's not my intention to put comments in at the last minute so they can't be responded to, and to the extent that comments can still be entered I'm happy to engage with them.
> > > >
> > > > In sum, my issue is that the paper makes broad claims that aren't sufficiently contextualized and which serve to run over existing thought on the (important!) issue raised without acknowledging it. That's not appropriate, and while "reject" could be seen as a strong reaction to that (specific, within the four corners of the paper) deficiency, nothing in the rebuttals has convinced me that there is willingness to fix it, so I continue to believe that a rejection is appropriate. Claims must be only as strong as the evidence that supports them, and here they over-reach. For example, the main claim of the responsibility schema is scoped to "all definitions of responsibility that we aim to consider", but the constraint in the claim is never actually specified, leaving the impression that it applies to all possible definitions, which just isn't true. When the overbreadth of the claims is pointed out, the response is that the claims are in fact narrow. Worse, the rebuttal contradicts itself, arguing both that the work is consonant with work in other communities on structural causation while heavily implying that structural causation should not be considered or that it can be equated to type causation. If the claims only apply to token causation, narrow the claims in the text. Here, for example, it would be appropriate to add a sentence or two early in the intro indicating that the claims are scoped only to token causation and that other notions of causation are explicitly being set aside. That's good! It makes the work stronger, if narrower.
> > > >
> > > > I continue to be confused by the unwillingness to engage the many arguments I've cited that describe causal and moral responsibility (and liability, where cases are generally more worked out because _lawyers have been thinking about this for many hundreds of years while logicians have only been trying to formalize it for about one hundred_) as *separate* constructs. I recognize that the proffered schema makes one a necessary antecedent of the other. This may be appropriate in token causation! I am not arguing that it isn't. What I am arguing is that the authors must engage with the many works I've cited (e.g., Hu, Kohler-Hausmann, Kasirzadeh and Smart) or explicitly limit the scope of their claims to the range which is defensible. Saying this is actually what was intended isn't enough: the claims themselves must be narrowed _in the paper_.
> > > >
> > > > Incidentally, I didn't actually say that the connection of the work to AI is weak - I don't think that at all, and was equally surprised by the claims of lack of relevance in other reviews. What I did say (not very clearly, unfortunately) is that there's a lot of existing work that attempts to _apply_ similar ideas to AI and finds that the important questions are largely about structural causation and that this renders a lot of counterfactual work less useful than its proponents claim. Not engaging with those arguments means that the (otherwise very interesting!) work reflects the same pitfalls as the work being critiqued in existing literature. The solution is to understand those arguments and build upon them - when is actual/token causation the relevant form of analysis? Why? Don't assume readers can fill in these baseline questions with answers.

---

> > > > > ### Author Response · Authors · 2023-08-22
> > > > >
> > > > > We thank the reviewer for further clarifying their position even past the deadline.
> > > > >
> > > > > _"the main claim of the responsibility schema is scoped to "all definitions of responsibility that we aim to consider", but the constraint in the claim is never actually specified, leaving the impression that it applies to all possible definitions, which just isn't true."_
> > > > >
> > > > > We fail to see how this impression can arise: we make explicit from the start that our focus lies with moral responsibility for particular outcomes, and we believe (in agreement with the literature) that our schema is universally applicable to that kind of responsibility. However, we will be more explicit on this point throughout the paper. We are also still unclear what the reviewer means when they question that the only form of causation relevant to responsibility for particular outcomes is that of actual causation, since that is per definition the notion of causation that concerns itself with particular outcomes. As to structural causation, we failed to find _any_ mention of "structural causation" in the work cited by the reviewer, so we are still unsure as to what the reviewer has in mind precisely with this terminology.
> > > > >
> > > > > "law": There is indeed hundreds of years of tradition in the law that studies the relation between causation and responsibility, and that tradition has consistently assumed, rather than questioned, that cause-in-fact (as actual causation is called within the law) is a necessary condition for the responsibility of outcomes. This is officially established through the _but for_ test, which is simply counterfactual dependence, but in practice legal experts are aware that this test is far too simplistic. In fact, the discrepancy between the but for test and intuitive judgments of causation was one of the main motivations for the entire literature on actual causation. (See [13] for details.) The deep connection between causation and responsibility is present already in the opening statement of the Standard Encyclopedia of Philosophy entry on Causation in the Law:
> > > > >
> > > > > "basic questions concerning causation in the law are: (i) what are the criteria in law for deciding whether one action or event has caused another (generally harmful) event; (ii) whether and to what extent causation in legal contexts differs from causation outside the law ... ; and (iii) what reason(s) (presumably _based in the law’s use of causation to attribute responsibility_) explain and/or justify such differences as may be found to exist. [emphasis added]"
> > > > >
> > > > > We firmly disagree with the reviewer that our work is in any way subject to the criticism voiced within the works of Hu, Kohler-Hausmann, and that of Kasirzadeh & Smart. We carefully looked at these articles, and they _exclusively_ criticize the use of counterfactuals involving _social categories such as race and gender_, and do so in the context of causal approaches to _fairness_ (and to a lesser extent, explainability). We do not focus on causal claims involving social categories such as race or gender, rather we focus on the actions of agents. Nor do we focus on fairness directly, although we imagine that a fully developed ethical framework for AI will have to integrate fairness into judgments about responsibility.
> > > > >
> > > > > We reiterate our agreement with the reviewer's claim that causation and responsibility are separate constructs, in two ways: causation is but one condition of responsibility, and only of one kind of responsibility, namely that of moral responsibility for particular outcomes. We also tried to show by citations from the work mentioned by the reviewer that our analysis actually seems entirely consistent it, and we are thus still unclear as to what criticism the reviewer has in mind.

---

### Decision · Program_Chairs · 2023-09-21

**Decision:**

Accept (poster)

**Comment:**

This paper offers a fresh perspective on the moral responsibility of AI systems, providing an interesting and clearly delineated new framework. Besides, it could benefit from a deeper exploration of practical applications and a more diverse philosophical backdrop. It has a good foundation, but branching out could really make it stand out. Overall, the meta-reviewer's recommendation is acceptance as a poster.